# North Atlantic oscillation controls multi-decadal changes in the North Tropical Atlantic−Pacific connection

Ruiqiang Ding [1,2] ✉, Hyacinth C. Nnamchi [3,4], Jin-Yi Yu [5], Tim Li [6], Cheng Sun [7], Jianping Li [8] ✉, Yu-Heng Tseng [9], Xichen Li [10], Fei Xie [7], Juan Feng[7], Kai Ji[1] & Xumin Li[1]

By exciting subtropical teleconnections, sea surface temperature (SST) anomalies in the North Tropical Atlantic (NTA) during boreal spring can trigger El Niño-Southern Oscillation (ENSO) events in the following boreal winter, thereby providing a precursor for ENSO predictability. However, this NTA−ENSO connection is not stationary, and it varies considerably over multi-decadal timescales, which cannot be directly explained by the Atlantic multi-decadal oscillation or the global warming trend. Here we show that multidecadal changes in the NTA−ENSO connection are principally controlled by multidecadal variability associated with the North Atlantic Oscillation (NAO). During the positive phase of the NAO, the amplification of the NTA impact on ENSO mainly arises from strengthening of the boreal spring mean precipitation over the equatorial Atlantic and enhancement of the persistence of NTA SST anomalies, which enhance the NTA influence by exciting stronger and more persistent subtropical teleconnections. Our findings show that multidecadal variability of the NAO is key to understanding the impacts of the NTA SST on the tropical Pacific Ocean.

El Niño-Southern Oscillation (ENSO), a fluctuation between unusually warm (El Niño) and cold (La Niña) conditions in the tropical Pacific, is the most prominent year-to-year climate variation on Earth[1]. ENSO exerts its profound impacts on remote regions of the globe through atmospheric teleconnections[2–4]. For example, ENSO is known to have a significant remote influence on sea surface temperature (SST) variability over the North Tropical Atlantic (NTA)[5–9], where significant warming in boreal spring occurs with a few months' lag following the peak phase of El Niño in winter. The NTA warming can force a

northward shift of the Atlantic intertropical convergence zone (ITCZ) with a subsequent impact on the frequency of Atlantic extreme hurricanes and precipitation in Northeast Brazil and Sahel[10–14]. Beyond the Atlantic sector, the NTA SST can exert pronounced impacts on the western North Pacific (WNP) tropical cyclone genesis[15], the WNP sub-tropical High[16], and the Indian Ocean Dipole[17]. In particular, the NTA SST may conversely play an important role in shaping the evolution of ENSO events and could therefore serve as a potential precursor for ENSO[18–22].

[1]State Key Laboratory of Earth Surface Processes and Resource Ecology, Beijing Normal University, Beijing, China. [2]Key Laboratory of Environmental Change and Natural Disasters of Chinese Ministry of Education, Beijing Normal University, Beijing, China. [3]GEOMAR Helmholtz Centre for Ocean Research Kiel, Kiel, Germany. [4]Department of Geography, University of Nigeria, Nsukka, Nigeria. [5]Department of Earth System Science, University of California, Irvine, CA 92697, USA. [6]Department of Atmospheric Sciences, University of Hawai'i at Manoa, Honolulu, HI, USA. [7]College of Global Change and Earth System Science (GCESS), Beijing Normal University, Beijing, China. [8]Frontiers Science Center for Deep Ocean Multispheres and Earth System (FDOMES)/Key Laboratory of Physical Oceanography/Institute for Advanced Ocean Studies, Ocean University of China, Qingdao, China. [9]Institute of Oceanography, National Taiwan University, Taipei, Taiwan. [10]Institute of Atmospheric Physics, Chinese Academy of Sciences, Beijing 100029, China. ✉e-mail: drq@bnu.edu.cn; ljp@ouc.edu.cn

The significance of the NTA precursor for ENSO depends on the stationarity of the NTA−ENSO connection. However, multidecadal variations in the NTA−ENSO relationship have been noted[23–25]. These variations have been attributed to changes in the Atlantic background state due to the Atlantic multidecadal oscillation (AMO)[23] or Atlantic secular warming trends[24,25]. Studies that have highlighted the importance of the AMO and Atlantic warming trends for changes in the NTA−ENSO connection are based on data since 1948. This period seems to be too short to properly measure the natural variability in the NTA−ENSO connection. Here we show that multidecadal variations in the NTA−ENSO connection cannot be directly explained by the AMO or the Atlantic warming trend when using data extending back to 1900; instead, multidecadal variations (21-yr running averages) of the North Atlantic Oscillation (NAO)[26,27] plays a key role in modulating the NTA−ENSO connection.

## Results

### Multidecadal variations in the NTA−ENSO connection

We first use the HadISST dataset during the period 1900–2021 to investigate multidecadal fluctuations in the NTA−ENSO connection. Figure 1a shows the sliding correlations on a 21-year moving window between the NTA SST index (see Methods) during boreal spring (March−May, MAM0, where "0" refers to the current year) and the Niño3.4 index during the subsequent winter (December−February, D0JF1, where "1" refers to the next year). The NTA−ENSO connection displays significant multidecadal changes, with statistically significant positive correlations for the periods 1913–1939 (hereafter P1) and 1993–2010 (hereafter P3), but no significant correlations for the period 1947–1985 (hereafter P2).

The correlation patterns of the MAM0 NTA SST index with the following D0JF1 tropical Pacific SST show a large region of significant negative correlations in the central and eastern equatorial Pacific during P1 and P3 and no regions with statistically significant negative correlations during P2 (Supplementary Fig. 1). Fluctuations in the NTA−ENSO connection are consistent across different SST datasets (Supplementary Fig. 2a), different lengths of the moving window (Supplementary Fig. 2b), and various ENSO indices (Supplementary Fig. 2c), indicating that observed multidecadal changes in the NTA−ENSO connection are robust.

### Modulations of the NTA−ENSO connection by the NAO

We have shown that the NTA−ENSO connection is not stationary, but varies considerably over multidecadal timescales. We next investigate the causes of the multidecadal fluctuations of the NTA−ENSO connection. Over multidecadal timescales, the SST variability over the North Atlantic basin is dominated by the AMO[28,29]. However, although the positive phase of the AMO coincides with the enhanced NTA−ENSO connection after the late 1990s[23], there is almost no simultaneous correlation between the AMO and multidecadal fluctuations of the NTA−ENSO connection over the entire period ($R = -0.15$; Supplementary Fig. 3). This suggests that multidecadal fluctuations of the NTA−ENSO relationship are unlikely to be directly explained by the AMO.

The NAO, a hemispheric meridional oscillation in atmospheric mass with centers of action near Iceland and over the subtropical Atlantic, displays irregular oscillations on interannual to multidecadal timescales[30,31]. In addition to ENSO, the NAO also exerts a strong influence on the northeasterly trades and SST over the NTA region[32–34]. Here we examine the possible linkage between the NAO and NTA−ENSO connection over multidecadal timescales. Figure 2 shows the composite differences in sea level pressure (SLP) and SST anomalies during the previous boreal winter−spring seasons (December−May, D−1JFMAM0) between P2 and P1 and between P3 and P2. Here, the winter−spring seasons were chosen because they are the seasons when the NTA SST anomalies typically develop and peak[35]. The SLP change from P1 to P2 is characterized by significant negative SLP anomalies in

the subtropical North Atlantic and significant positive anomalies in the subpolar North Atlantic, which indicates a negative phase of the NAO[26,27] (Fig. 2a). In contrast, the SLP change from P2 to P3 is featured by significant positive and negative SLP anomalies in the subtropical and subpolar North Atlantic, respectively, which indicates a positive phase of the NAO (Fig. 2b). Consistent with the SLP changes, both the

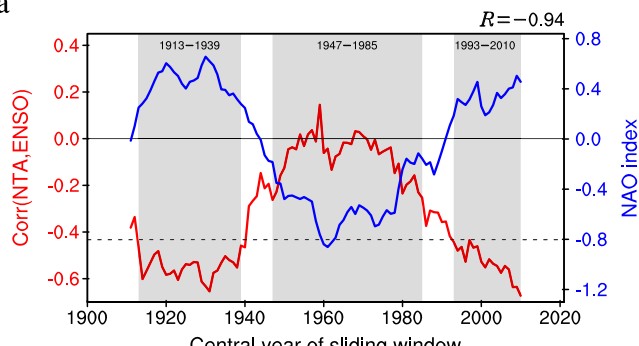

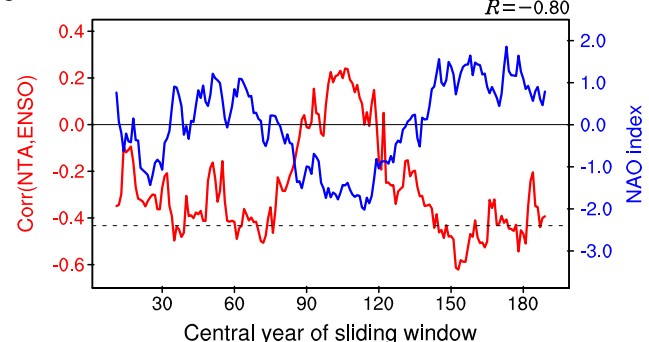

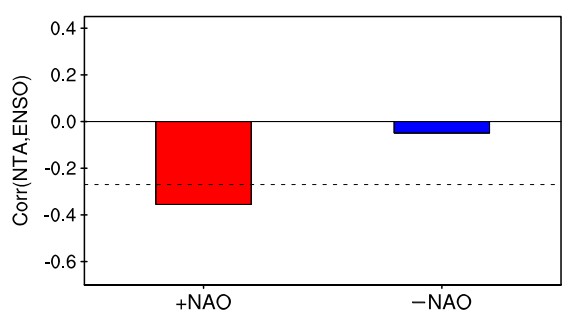

**Fig. 1 | Relationship between the North Tropical Atlantic (NTA)−El Niño-Southern Oscillation (ENSO) connection and North Atlantic Oscillation (NAO) in observations and models. a** The 21-year sliding correlation coefficients between the boreal spring (MAM0) NTA sea surface temperature (SST) index and subsequent boreal winter (D0JF1) Niño3.4 index (red line), and the 21-year running averages of the normalized NAO index during the previous boreal winter−spring seasons (D−1JFMAM0) (blue line) for the period 1900–2021 in observations. The correlation coefficient between the NTA−ENSO connection and NAO is −0.94 (significant at the 95% confidence level). The impact of the previous winter (D−1JF0) ENSO has been removed from the MAM0 NTA SST index using linear regression with respect to the Niño3.4 index, and the dashed line indicates the 95% confidence level for the sliding correlation coefficients. Bands of gray shading indicate two high correlation periods (1913–1939 and 1993–2010) and a low correlation period (1947–1985) of the NTA−ENSO connection. **b** Same as (**a**), but for the NTA−ENSO connection and NAO derived from the 200-yr pre-industrial model simulation by the Community Earth System Model (CESM). The correlation coefficient between the modeled NTA−ENSO connection and NAO is −0.80 (significant at the 95% confidence level). **c** The correlation coefficients between the spring NTA SST index and subsequent winter Niño3.4 index in the positive-NAO and negative-NAO forcing experiments. The dashed line denotes the 95% confidence level.

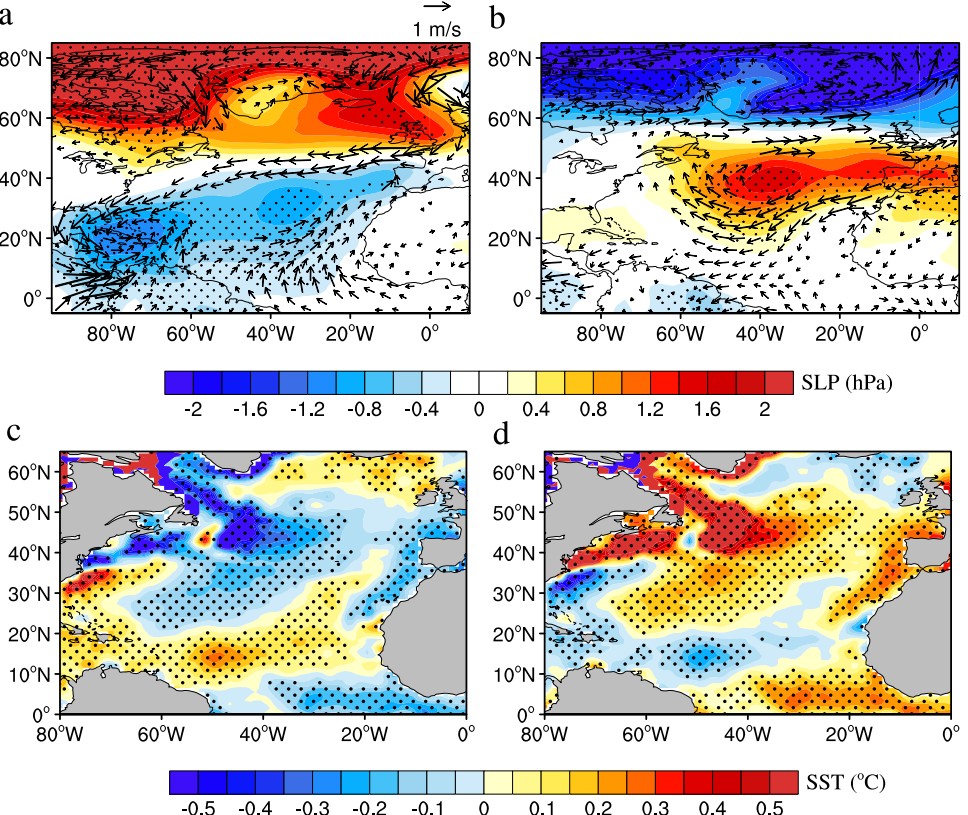

**Fig. 2 | Composite differences of sea level pressure (SLP), 850-hPa winds, and sea surface temperature (SST) anomalies. a, b** Composite differences of 850-hPa winds (vectors, m s⁻¹) and SLP (shading, hPa) anomalies during the previous boreal winter−spring seasons (D−1JFMAM0) between 1947–1985 (P2) and 1913–1939 (P1) (**a**) and between 1993–2010 (P3) and 1947–1985 (P2) (**b**). **c, d** As in **a, b**, but for composite differences of D−1JFMAM0-averaged SST (shading, °C) anomalies between P2 and P1 (**c**) and between P3 and P2 (**d**). In **a, b**, only 850-hPa wind anomalies significant at the 95% confidence level are shown. In **a–d**, stipples indicate SLP and SST anomalies significant at the 95% confidence level.

SST changes from P1 to P2 and from P2 to P3 feature a tripole structure in the North Atlantic associated with the NAO (Fig. 2c,d), rather than the basin-wide warming or cooling pattern associated with the AMO[28,29].

Multidecadal fluctuations in the NTA−ENSO connection are strongly out of phase with those of the D−1JFMAM0-mean NAO index ($R = -0.94$; Fig. 1a). In addition, the analysis of a 200-year pre-industrial control simulation of the Community Earth System Model (CESM) (see Methods), with more NAO cycles, also support our observational results. Multidecadal variations of the modeled NTA−ENSO connection are significantly negatively correlated with those of the modeled NAO ($R = -0.80$) (Fig. 1b).

Two sets of ensemble experiments were further performed using the CESM to demonstrate that the NTA−ENSO connection can indeed change with the phases of the NAO forcing (see Methods). In the positive-NAO ensemble run, there is a significant negative correlation between the MAM0 NTA SST index and the following D0JF1 Niño3.4 index (Fig. 1c). In contrast, in the negative-NAO ensemble run, the significant negative correlation disappears. These modeling results support the observational analysis, indicating that multidecadal variability in the NTA−ENSO connection and NAO are indeed anti-correlated.

**Physical mechanisms driving the NAO's modulation effect**
Previous studies have proposed a subtropical teleconnection mechanism (STM) whereby the NTA SST could impact ENSO[18–20,36]. The STM supposes that the boreal spring NTA warming enhances convection over the Atlantic ITCZ to excite a cyclonic flow over the subtropical eastern Pacific through a Gill-type Rossby wave response, which then gives rise to an anticyclonic flow over the subtropical western−central Pacific through interactions with the Pacific ITCZ precipitation. This anticyclonic flow generates easterly wind anomalies over the equatorial western Pacific that initiate surface cooling over the equatorial central Pacific a few months later.

Here we show that the STM operates only when the NAO is in its positive phases (P1 and P3) but not in its negative phase (P2). The NTA warming is able (unable) to excite a pair of low-level circulation anomalies over the subtropical Pacific from boreal spring to summer during P1 and P3 (during P2), which is favorable (less favorable) for the development of surface cooling in the equatorial Pacific during the subsequent seasons (Fig. 3). In addition to the subtropical pathway, the NTA SST can also impact ENSO evolution by inducing eastward-propagating Kelvin waves (i.e., via the equatorial pathway)[18,22]. Easterly wind anomalies occur from the Indian Ocean to the western Pacific in boreal summer during P3 (Fig. 3j), which might result from a Kelvin-wave response to the equatorial diabatic heating[37] associated with the NTA SST.

We now investigate the possible mechanisms responsible for the NAO modulation of the subtropical teleconnections associated with the spring NTA SST. We propose two possible mechanisms to explain the NAO's modulation effect. The first mechanism involves a direct modulation of the mean precipitation over the equatorial Atlantic. The positive phase of the NAO strengthens the northeasterly trade winds during boreal spring, thereby promoting low-level convergence over the equatorial Atlantic (Supplementary Fig. 4a) and enhancing precipitation and relative humidity in the lower troposphere there

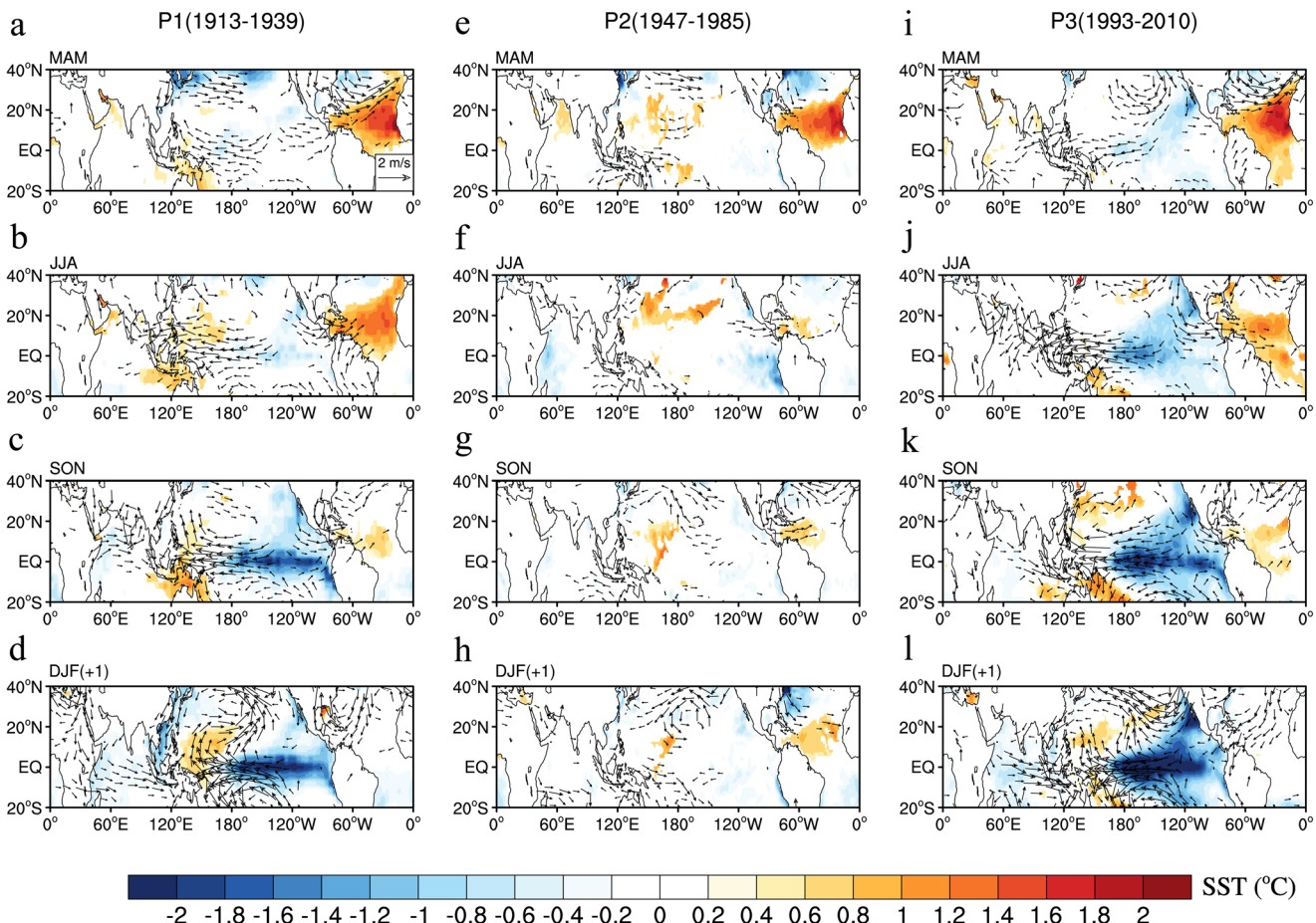

**Fig. 3 | Regressions with respect to the spring North Tropical Atlantic (NTA) sea surface temperature (SST). a–d** Regressions of SST (shading) and 850-hPa winds (vectors) with respect to the MAM0 NTA SST index during 1913–1939 (P1) for MAM (**a**), JJA0 (**b**), SON0 (**c**), and D0JF1 (**d**) seasons. **e–h** As in **a–d**, but during 1947–1985 (P2) for MAM0 (**e**), JJA0 (**f**), SON0 (**g**), and D0JF1 (**h**) seasons. **i–l** As in **a–d**, but during 1993–2010 (P3) for MAM0 (**i**), JJA0 (**j**), SON0 (**k**), and D0JF1 (**l**) seasons. The impact of the previous winter (D−1JF0) El Niño-Southern Oscillation (ENSO) has been removed from the MAM0 NTA SST index using linear regression with respect to the Niño3.4 index. Only 850-hPa winds and SST anomalies significant at the 95% confidence level are shown.

(Supplementary Fig. 4b,c). The wetter basic state over the equatorial Atlantic in the positive phase of the NAO can lead to stronger local precipitation responses to NTA SST anomalies[38,39] (see Methods; Supplementary Figs. 5 and 6a), thereby exciting stronger subtropical teleconnections that are more effective in relaying the Atlantic influences into the tropical Pacific.

The second mechanism involves modulation of the NTA SST persistence. In the positive (negative) phase of the NAO, the ocean mixed layer deepens (shoals) over the NTA region in response to a strengthening (weakening) of the northeasterly trade winds (Supplementary Fig. 7). As a result of the deeper (shallower) mixed layer, thermal anomalies stored in the mixed layer would persist longer (shorter)[40,41]. In addition, the intensified northeasterly trade winds tend to enhance the wind–evaporation–SST (WES) feedback mechanism[42,43] (see Methods), which acts to increase the persistence of the NTA SST anomalies. The NTA SST anomalies decay faster from spring to summer in the negative phase of the NAO than in the positive phase of the NAO for both the observations (Fig. 3) and models (Supplementary Figs. 5 and 6b). Multidecadal variations in the persistence of the NTA SST anomalies from spring to summer (see Methods) are positively and negatively correlated with those of the NAO phase and NTA−ENSO connection, respectively ($R = 0.80$ and $-0.81$, both significant at the 90% confidence level; Supplementary Fig. 8).

To understand how variations in the NTA SST persistence affect the NTA-induced subtropical teleconnections[18–20], we first perform two

sensitivity experiments using an atmospheric general circulation model (AGCM) to examine the atmospheric responses to both the spring NTA SST anomalies and their component that persists into summer (termed the spring-to-summer NTA SST) (see Methods; Fig. 4a,b). The spring NTA SST-induced precipitation anomalies occur mainly over the equatorial Atlantic where the Atlantic ITCZ is located (Fig. 4c). The associated diabatic heating produces a low-level cyclonic flow over the subtropical eastern Pacific, consistent with the observations[18] (Fig. 3). In contrast, along with the northward migration of the ITCZ during boreal summer, the spring-to-summer NTA SST-induced precipitation anomalies are slightly shifted to the north of the equatorial Atlantic and expand westward into the Caribbean Sea (Fig. 4d), and so become closer to the subtropical eastern Pacific. Therefore, although the amplitude of the spring-to-summer NTA SST is generally weaker than that of the spring NTA SST (Fig. 4a,b), the cyclonic circulation response over the subtropical eastern Pacific to the spring-to-summer NTA SST seems stronger than that to the spring NTA SST (Fig. 4e,f). This suggests that despite the weaker amplitude, the spring-to-summer NTA SST can produce robust teleconnections over the subtropical eastern Pacific and therefore plays a crucial role in effectively and continuously relaying the Atlantic influences into the Pacific.

The importance of the NTA SST persistence in forcing ENSO variability is further substantiated by coupled model sensitivity experiments with prescribed spring SST anomalies of varying

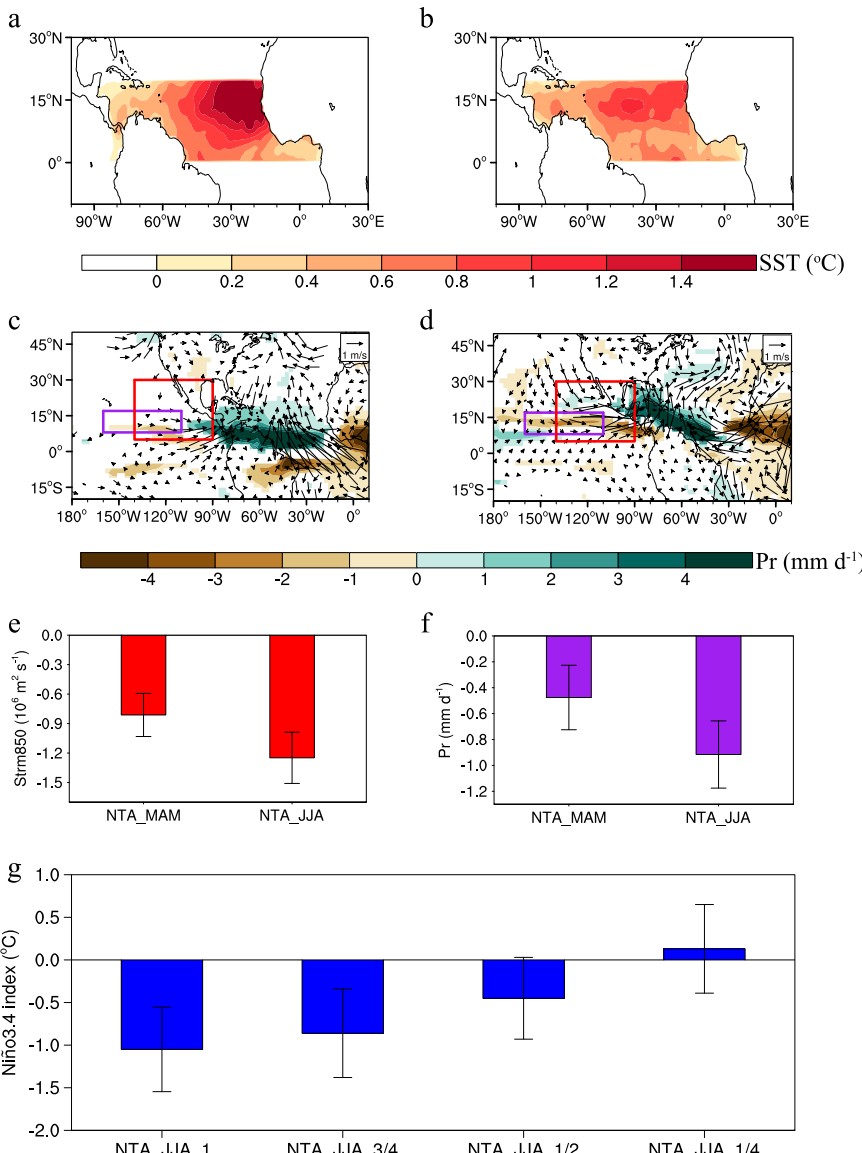

**Fig. 4 | Simulated impacts of the North Tropical Atlantic (NTA) sea surface temperature (SST) persistence on the NTA−El Niño-Southern Oscillation (ENSO) connection. a, b** Imposed spring (**a**) and summer (**b**) NTA SST forcings (see Methods) in the atmospheric general circulation model (AGCM) experiments. **c, d** Responses of 850-hPa winds (vectors; m s⁻¹) and precipitation (shaded; mm d⁻¹) anomalies to the spring (**c**) and summer (**d**) NTA SST forcings in the AGCM experiments. Only 850-hPa winds and precipitation anomalies significant at the 95% confidence level are shown. **e** 850-hPa stream function anomalies area-averaged over the region (5°–30°N, 140°–90°W) induced by spring and summer NTA SST

forcings. **f** As in **e**, but for precipitation anomalies area-averaged over the region (8°–17°N, 160°–110°W) induced by spring and summer NTA SST forcings. **g** Responses of the tropical Pacific Ocean to summer NTA SST anomalies of different magnitudes in the coupled general circulation model (CGCM) experiments (see Supplementary Fig. 9 for imposed summer NTA SST anomalies of different magnitudes). In **c**, **d**, the red (purple) boxes represent the domain used to compute the area-averaged 850-hPa stream function (precipitation) anomalies. In **e**–**g**, error bars indicate the 95% confidence intervals.

magnitudes (Supplementary Fig. 9). We find that the simulation with a greater magnitude of summer NTA SST anomalies produces a stronger La Niña signal over the equatorial Pacific during the following winter (Fig. 4g), consistent with the recent findings of Jiang et al.[44].

Figure 5 finally shows a schematic diagram summarizing the physical mechanisms through which the NAO modulates the NTA SST−ENSO connection. When the boreal winter–spring NAO is in its positive phase, this leads to stronger-than-average mean northeasterly trade winds over the NTA region[34]. On one hand, the intensified northeasterly trade winds enhance the low-level convergence and favor stronger mean precipitation over the equatorial Atlantic during boreal spring, which provides favorable conditions for the NTA SST to excite stronger subtropical teleconnections (Fig. 5a). On the other

hand, the intensified northeasterly trade winds enhance the persistence of the spring NTA SST anomalies, which helps maintain the subtropical teleconnections that continuously relay the Atlantic influences into the tropical Pacific (Fig. 5b). The combination of these two processes may ultimately lead to amplified NTA impacts in the tropical Pacific.

## A likely weakening of the NTA−ENSO connection in the near future

Although multidecadal variability in the NAO contributes significantly to multidecadal fluctuations in the NTA−ENSO connection, the origin of this variability remains unclear. Previous studies have reported that a multidecadal feedback loop may exist between the NAO, AMO, and

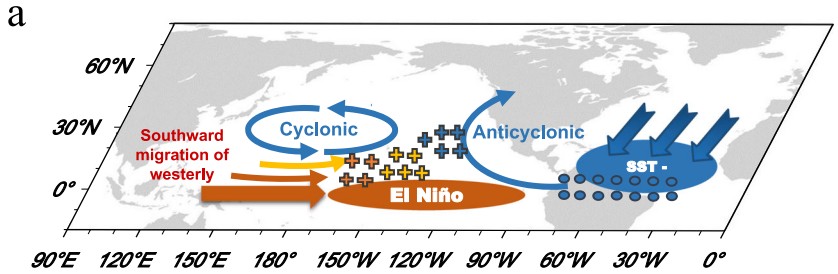

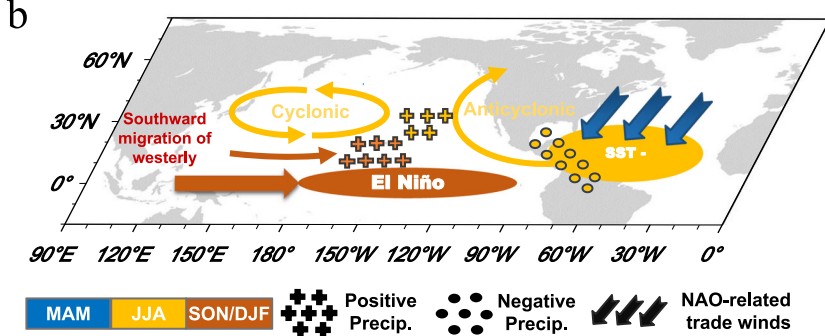

**Fig. 5 | Schematic representation of two major mechanisms behind the North Atlantic Oscillation (NAO) modulation of the North Tropical Atlantic (NTA)−El Niño-Southern Oscillation (ENSO) connection (motivated by Fig. 3 in Kug et al.[20]). a** NAO modulation of background mean-state precipitation over the equatorial Atlantic. The intensified northeasterly trade winds associated with the positive NAO favor stronger background mean-state precipitation over the equatorial Atlantic during boreal spring. The wetter basic state would lead to a stronger negative precipitation anomaly response to a negative NTA sea surface temperature (SST) anomaly (SST−), thereby exciting stronger subtropical teleconnections that more readily relay the Atlantic influences into the tropical Pacific. **b** NAO modulation effect on NTA SST persistence. The intensified northeasterly trade winds associated with the positive NAO deepen the ocean mixed layer and enhance the wind−evaporation−SST (WES) feedback mechanism, which in turn increase the persistence of the NTA SST anomalies from spring to summer, thereby generating more persistent subtropical teleconnections that continuously relay the Atlantic influences into the tropical Pacific. Colors denote the seasons when significant anomalies occur. Crosses and circles denote the locations of positive and negative precipitation anomalies, respectively.

Atlantic Meridional Overturning Circulation (AMOC)[45–50]. On multi-decadal timescales, significant negative correlations are found when the NAO lags the AMO by 18–23 years (Supplementary Fig. 10a). This implies that despite the lack of simultaneous correlation between the AMO and NTA−ENSO connection, the AMO could exert an indirect influence on the NTA−ENSO connection through its interaction with the NAO. The lead-lag relationship between the NAO and AMO on multidecadal timescales provides a simple way to predict the NAO around two decades in advance. Therefore, we developed an AMO-based linear model for predicting multidecadal NAO index (see Methods). The model can capture the NAO phase shift in recent decades (Supplementary Fig. 10b). The model prediction suggests that the NAO phase will experience a shift from positive to negative in the near future (before 2030). Given that the NTA−ENSO connection is closely connected to the NAO phase, the NAO phase reversal from positive to negative would lead to a weakening of the NTA−ENSO connection in the coming decade.

## Discussion

We have shown that the multidecadal variations in the NTA−ENSO connection are primarily controlled by the NAO multidecadal variability. These findings have far-reaching implications for ENSO prediction and a better understanding of inter-basin interactions between the Atlantic and Pacific Oceans.

Our intention is to emphasize that the NTA−ENSO connection is primarily controlled by multidecadal climate variability associated with the NAO, but we cannot entirely rule out the possibility that the warmer background state in the NTA region due to the global warming trend could be contributing to the enhanced NTA−ENSO relationship after the early 1990s[23–25]. However, the warmer background state in the NTA region may have both positive and negative effects on the NTA−ENSO connection. On one hand, enhanced convection and precipitation due to warmer SSTs in the NTA region are more favorable for relaying the Atlantic influences into the tropical Pacific[23]; however, on the other hand, a faster decay of the NTA SST anomalies in a warmer climate, due to enhanced evaporation and reduced shortwave radiation[51], may also contribute to a weakened NTA−ENSO connection. This implies that influences of the warmer background states on the NTA−ENSO connection remain elusive, possibly depending on whether the positive effects outweigh the negative ones, or the opposite. The relative importance of natural variability associated with the NAO and global warming trends should be further explored to achieve a comprehensive understanding of the causes for variations in the NTA−ENSO connection.

It has been well recognized that two types of El Niño exist in the tropical Pacific, characterized by larger SST anomalies over the eastern Pacific (EP) and central Pacific (CP), respectively[52–55]. The NTA SST has been suggested to be closely related to the development of the CP El Niño[18]. Moreover, it may also promote a faster ENSO phase transition, thus helping to shape the dominant period of ENSO[18,23]. Our findings further indicate that the NAO may influence the ENSO properties (including its spatial pattern and period) by modulating the NTA−ENSO connection. The weakening of the NTA−ENSO connection in the near future would likely lead to changes in ENSO properties. In addition, Chen et al.[56] suggested that the North Pacific Oscillation (NPO) could also have a modulation influence on the NTA−ENSO connection. It is likely that the NAO and NPO may play a joint role in modulating the NTA−ENSO connection. Further investigations are required to better understand these two important issues.

## Methods

### Observation-based data

Four monthly SST datasets were used in this study: (1) the Hadley Center Sea Ice and SST dataset version 3 (HadISST; 1871–2021)[57]; (2) the National Oceanic and Atmospheric Administration Extended Reconstructed SST version 4 (ERSST; 1854–2021)[58]; (3) the Kaplan Extended SST version 2 (Kaplan SST; 1856–2021)[59]; and (4) the Centennial in situ Observation-Based Estimates SST version 2.9.2 (COBE SST; 1850–2021)[60]. Monthly tropospheric winds and SLP fields were derived from the "merged" NCEP atmospheric reanalysis dataset. The merged NCEP dataset was produced from the NOAA-CIRES Twentieth Century Reanalysis (20CRv2c; 1851–2014)[61] and the NCEP-DOE Reanalysis 2 (1979–2021)[62]. To ensure temporal consistency, we use monthly climatological differences between the 20CRv2c and NCEP/DOE2 data during the 1979–2014 overlap period to calibrate the mean state of NCEP/DOE2. The precipitation and relative humidity data used were from the NCEP/NCAR Reanalysis 1 (1948–2021)[63]. The monthly ocean mixed layer depth (MLD) dataset was taken from the ECMWF Ocean Reanalysis System 5 (ORAS5; 1975–2021)[64], which is a global eddy-permitting ocean-sea ice reanalysis system.

Because of relatively large uncertainties in surface observations prior to 1900, we limit our analysis to datasets from 1900 onward. All of the anomalies are calculated by first removing the average seasonal cycle over the period 1981–2010 and then removing the long-term linear trend using the least squares method.

### The 200-yr pre-industrial simulations of the CESM

Given the limited record of SST and SLP observations, we further examined the relationship between multidecadal fluctuations of the NTA−ENSO connection and NAO in a long (200-yr) pre-industrial simulation from the Community Earth System Model (CESM) developed at the National Center for Atmospheric Research (NCAR)[65].

### Climate indices

The NTA SST index is defined as the average of SST anomalies over the NTA region (0°–15°N, 90°W–20°E)[18]. The Niño3.4 index, the most commonly used index to represent ENSO intensity, is defined as the average of SST anomalies over the Niño3.4 region (5°S–5°N, 170°W–120°W). The AMO index is calculated as the detrended SST anomalies averaged over the North Atlantic region (0°–60°N, 80°W–0°)[66]. The NAO index is defined as the principal component (PC) time series of the leading Empirical Orthogonal Function (EOF) of SLP anomalies over the Atlantic sector (20°–80°N, 90°W–40°E)[32].

### Moisture budget

From the moisture budget perspective, on the interannual variability, tropical precipitation anomalies are mainly dominated by the moisture convergence induced by anomalous circulation[67]:

$$P' \sim - <\bar{q} * \nabla \cdot \vec{u}'>, \quad (1)$$

where $P$ denotes the tropical precipitation, the prime denotes interannual variability, $\bar{q}$ denotes the background mean-state specific humidity, and $\nabla \cdot \vec{u}'$ denotes the moisture convergence induced by anomalous circulation, respectively. <> denotes vertical integration. For the same magnitude of anomalous convergence induced by local SST forcing, the precipitation response would be stronger when the mean-state low-level moisture were more. Thus, the wetter basic state over the equatorial Atlantic in the positive phase of the NAO could lead to stronger local precipitation responses to NTA SST anomalies.

### The WES feedback parameter

Following Czaja et al.[33] and Vimont et al.[68], the sensitivity of latent heat flux to zonal winds, or the WES parameter (WESp) can be expressed by

the following relationship:

$$WESp = -\frac{\partial LH}{\partial u} = -LH\frac{u}{W^2}, \quad (2)$$

where LH is latent heat flux, $u$ is the 10 m zonal wind, and $W$ is the total wind speed. The total wind speed can be decomposed as:

$$W = \sqrt{u^2 + v^2 + w_*^2}, \quad (3)$$

where $w_*$ is turbulent background wind speed. Since zonal wind dominates the WES feedback, Eq. (2) can be simply expressed by:

$$WESp \sim \frac{1}{1 + \left(\frac{w_*}{u}\right)^2}. \quad (4)$$

Equation (4) indicates that a stronger mean zonal wind generally corresponds to a more intense WES feedback, and vice versa. The intensified northeasterly trade winds are associated with a stronger mean zonal wind, which tend to enhance the WES feedback mechanism[43].

### Spring-to-summer persistence of the NTA SST

Autocorrelation analysis was used to measure the spring-to-summer persistence of the NTA SST, which is defined as the correlation between the time series of the boreal spring NTA SST index and the time series of the following summer NTA SST index over a period.

### An AMO-based linear model for predicting multidecadal NAO

The lead-lag relationship between the NAO and AMO on multidecadal timescales provides a simple way to predict the NAO around two decades in advance. Namely, the AMO index being shifted by 21 years can serve to predict the NAO index on multidecadal timescales. Therefore, an AMO-based linear model is developed to predict multidecadal NAO:

$$NAO(t) = a * AMO(t - 21) + b. \quad (5)$$

where $t$ is time in years, and NAO and AMO represent the boreal winter−spring (DJFMAM) NAO and AMO indices, respectively. The coefficients $a$ and $b$ are determined using the least squares regression method based on the data over the historical period (1932–2010). The reason for starting in 1932 is that the original data, which started in 1900, became available from 1911 after a 21-year sliding average. The linear model requires the use of the AMO index 21 years ahead, so the training period of the linear model started in 1932. Based on the preceding AMO index, we used the linear model to carry out a prediction for multidecadal NAO in 2011–2031.

### Significance tests

The statistical significance of the correlations and regressions was determined using a two-tailed Student's $t$-test. To account for the temporal autocorrelation of the time series, the number of effective degrees of freedom was estimated using the method of Bretherton et al.[69]. We used a bootstrap method[70] to examine whether the NTA−ENSO connections for positive and negative NAO phases are significantly different. Bootstrapping is a resampling method used to generate samples from a dataset using the replacement technique.

### Climate model experiments

To examine the atmospheric responses to spring and summer NTA SST forcings, we performed atmospheric general circulation model (AGCM) experiments using Atmospheric Model version 2.1 (AM2.1) from the Geophysical Fluid Dynamics Laboratory (GFDL)[71]. The model horizontal resolution is 2.5° longitude × 2° latitude, with 24 vertical

levels. We performed two sets of AGCM experiments, forced with spring and summer NTA SST anomalies, respectively. The imposed spring (summer) NTA SST forcing was obtained by regressing spring (summer) SST anomalies over the NTA region onto the spring NTA SST index (Fig. 4a,b). Outside the NTA region, SST was set to monthly climatology. The model was integrated for 30 years, with the first 10 years discarded as spin-up and the last 20 years used for analysis.

To verify the modulation effect of the NAO phase on the NTA−ENSO connection, we performed coupled general circulation model (CGCM) experiments using the CESM. The positive-NAO run is generated by forcing the model with wind stress anomalies associated with the positive phase of the NAO (Supplementary Fig. 11a) superimposed on the climatological wind stress fields during the controlled flux stage from 1 December(−1) to 31 May(0) (Supplementary Fig. 11b). The fully coupled stage then begins on 1 June(0), after which the model is integrated for the next 9 months until 28 February(1). For the negative-NAO run, the anomaly forcing is simply reversed in sign. This modeling scheme is adapted from Chakravorty et al.[72]. The positive-NAO and negative-NAO experiments consisted of 60-member ensemble simulations from the control run, each initialized with conditions on 1 December from the last 60 years of the 80-year integration.

To highlight the key role of the NTA SST persistence in the NTA impact on ENSO, we also performed four sets of CGCM experiments using the CESM. In four sets of model experiments, spring NTA SST anomalies were fixed as regressions of spring SST anomalies with respect to the concurrent NTA SST index for the period 1979–2021, but the magnitude of summer NTA SST anomalies was set to 1.0, 3/4, 1/2, and 1/4 of the magnitude of the prescribed summer NTA SST anomalies (that is, regressions of summer SST anomalies with respect to the preceding spring NTA SST index). Supplementary Fig. 9 shows the prescribed SST anomalies used in four sets of sensitivity experiments. Each set of sensitivity experiments consisted of 60-member ensemble simulations with different initial conditions.

## Data availability

The data that support the findings of this study are freely available. The HadISST dataset is available at http://www.metoffice.gov.uk/hadobs/hadsst3/. The ERSST dataset is available at https://psl.noaa.gov/data/gridded/data.noaa.ersst.v4.html. The Kaplan SST dataset is available at https://psl.noaa.gov/data/gridded/data.kaplan_sst.html. The COBE SST dataset is available at https://psl.noaa.gov/data/gridded/data.cobe2.html. The 20CRv2c dataset is available at https://psl.noaa.gov/data/gridded/data.20thC_ReanV2c.html. The NCEP-DOE Reanalysis 2 is available at https://psl.noaa.gov/data/gridded/data.ncep.reanalysis2.html. The NCEP/NCAR Reanalysis 1 is available at https://psl.noaa.gov/data/gridded/data.ncep.reanalysis.html. The ORAS5 dataset is available at https://www.ecmwf.int/en/forecasts/charts/oras5/.

## Code availability

The data in this study were analyzed with NCAR Command Language (NCL; http://www.ncl.ucar.edu/). All relevant codes used in this study are available, upon request, from the corresponding author R.Q.D.

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

## Acknowledgements

This research was jointly supported by the National Natural Science Foundation of China (42225501, 41975070), and China's National Key Research and Development Projects (2020YFA0608402). Y.-H.T. was supported by MOST Grant# 107-2611-M-002-013-MY4 "Improving the Decadal Climate Prediction using a New Fully-Coupled Global Climate System". H.C.N. was supported by the Deutsche Forschungs Gemeinschaft (DFG) project "NOVEL" through grant 456490637.

## Author contributions

R.Q.D. and J.P.L. designed the study. R.Q.D. wrote the paper. K.J. and X.M.L. performed the data analysis and prepared all figures. X.M.L. conducted the modeling experiments. H.C.N., T.L., J.-Y.Y., Y.-H.T., C.S., X.C.L., J.F., and F.X. contributed to the interpretation of the results and the improvement of the manuscript.

## Competing interests

The authors declare no competing interests.
