## [Peer Review File · Nature Communications]

North Atlantic Oscillation controls multidecadal changes in the North Tropical Atlantic–Pacific connectionREVIEWER COMMENTS

Reviewer #1 (Remarks to the Author):

This study provides a new explanation for the multidecadal changes in the North Tropical Atlantic (NTA)–ENSO connection, suggesting that the multidecadal variability associated the North Atlantic Oscillation (NAO) could principally control the multidecadal changes in the NTA–ENSO connection. The findings show that the phase of the NAO is key to understanding the impacts of the NTA SST on the tropical Pacific Ocean. This manuscript presents a novel view of tropical Atlantic-Pacific inter-basin interactions, results of which have important implications for ENSO prediction and a better understanding of inter-basin interactions between the tropical Atlantic and Pacific. These results presented in this manuscript are interesting. This manuscript is well-written and the obtained results are reasonable. This manuscript is suggested to be accepted for publication in the journal after some revisions.

1. The springtime NTA SST may have important climatic effects in many aspects. As stated in the introduction of the manuscript, the NTA SST may play an important role in shaping the evolution of ENSO events and could therefore offer a potential precursor for ENSO. In addition to the potential precursor for ENSO, the springtime NTA SST has also been suggested to play important roles in modulating the western North Pacific (WNP) tropical cyclone (TC) genesis (Huo et al., 2015), western Pacific subtropical High (Wang et al., 2017; Chen et al., 2021), and the Indian Ocean Dipole (Zhang et al., 2022). Therefore, the springtime NTA SST may also offer a potential precursor for WNP TC numbers, the intensity of the western Pacific subtropical High, and the Indian Ocean Dipole. These associated information are suggested to be added in the manuscript, to further highlight the importance of NTA SST in climate variability and climate prediction.

Huo L, Guo P, Hameed S N, et al. The role of tropical Atlantic SST anomalies in modulating western North Pacific tropical cyclone genesis[J]. Geophysical Research Letters, 2015,42(7): 2378-2384.

Wang L, Yu J Y, Paek H. Enhanced biennial variability in the Pacific due to Atlantic capacitor effect[J]. Nature communications, 2017,8(1): 1-7.

Chen S, Chen W, Wu R, et al. Performance of the IPCC AR6 models in simulating the relation of the western North Pacific subtropical high to the spring northern tropical Atlantic SST[J]. International Journal of Climatology, 2021,41(4): 2189-2208.

Zhang G, Wang X, Xie Q, et al. Strengthening impacts of spring sea surface temperature in the north tropical Atlantic on Indian Ocean dipole after the mid-1980s[J]. Climate Dynamics, 2022: 1-16.

2. One recent study (Chen et al. 2022) proposed that the preceding winter North Pacific Oscillation (NPO) is an important source of the internal climate variability that modulates the spring NTA–ENSO connection. However, the observation data used in Chen et al. (2022) is from 1948 and is shorter than the data used in this study. It is suggested to further verify whether NPO or NAO control the multidecadal changes in the NTA–ENSO connection? Or both NPO and NAO may make contributions. Any linkage between the multidecadal variability of the NAO and the NPO?

Chen S, Chen W, Yu B, et al. Impact of internal climate variability on the relationship between spring northern tropical Atlantic SST anomalies and succedent winter ENSO: the role of the North Pacific Oscillation[J]. Journal of Climate, 2022,35(2): 537-559.

3. The changes in the NTA–ENSO connection may influence the ENSO properties largely. It is suggested to add more relevant discussions in this regard in the manuscript. Considering that the NTA-triggered ENSO tends to be the central Pacific (CP) type rather than the eastern Pacific (EP) type, the multidecadal changes in the NTA–ENSO connection may influence the dominant degree of EP and CP ENSO in different decades. In addition, the NTA SST could promote a faster ENSO phase transition, thus helping to shape the dominant period of the ENSO (Is the quasi-biennial (~2-3 yr) rhythm or the

quadrennial ($\sim 4-5$ yr) rhythm dominant?) As stated in the manuscript, the AMO-based linear model predictions suggest a likely weakening of the NTA-ENSO connection in the near future. How this weakening of the NTA-ENSO connection may influence the ENSO properties (the dominant type and the dominant period) in the near future? These points may further highlight the importance of the NTA-ENSO connection changes in the climate system.

4. Lines 177-180: "The intensified northeasterly trade winds can enhance the wind-evaporation-SST (WES) feedback mechanism, which acts to increase the persistence of the NTA SST anomalies. " Here, could the enhancement of the WES feedback mechanism be supported by the observational data? Why the intensified northeasterly trade winds can enhance the WES feedback? Why these mentioned changes could act to increase the persistence of the NTA SST anomalies? Any local air-sea positive feedback is included in the process?

Minor comments:

1. Line 248 : "inter-basin interactions between the Atlantic". Some words are missing here: inter-basin interactions between the Atlantic and the Pacific ?

2. Line 24: Nature Geoscience should be Nature Communications.

3. For the title, "the multidecadal changes" need to be included? Such as, North Atlantic Oscillation controls the multidecadal changes in the North Tropical Atlantic-Pacific connection.

Reviewer #2 (Remarks to the Author):

The manuscript "North Atlantic Oscillation controls the North Tropical Atlantic-Pacific connection" by Ruiqiang Ding et al deals with the north tropical Atlantic impact on ENSO and its non-stationarity. The authors suggest that the decadal variability of the North Atlantic Oscillation acts as modulator of this connection, being during the positive phase of NAO when the connection works.

The authors indicate that, although the positive phase of AMO coincides with enhancement of NTA-ENSO, there is almost no correlation between the correlation curve NTA-ENSO and AMO but yes with that of the NAO .

Although the experimental set up is fine and also the results seem to be promising, I have some concerns about the explanations given in the paper and, for this reason, I cannot recommend this paper to be published in its present form in this high impact journal.

The linear model based on AMO implies that positive AMO produces, 20 year later a positive NAO. What is the mechanism behind? On the one hand, the authors use a linear model to predict NAO as a function of AMV , but NAO can be internally. Regardless of that, I am concern about the signs of the relations found. For example, Sutton et al (2019) indicates, in its figure 8, how a positive AMV phase associated with a stronger AMOC leads to a negative winter NAO response (i.e., weaker subtropical high and subpolar low over the North Atlantic) , so NAO- is related to positive AMV. Nevertheless, in the present manuscript the authors obtain a positive NAO with NTA-ENSO relation, which is not in agreement with the results from Wang et al (2017) in which the authors relate NTA-ENSO with positive AMV. Even if NAO can be predicted from AMV, the sign of the relation does not agree with Wang et al (2017) results. Please clarify this.

Also, there are some parts of the text that are very difficult to read. For example, the authors claim that the positive NAO strengthen the northeasterly winds promoting low level convergence over the equatorial Atlantic and enhancing precipitation and relative humidity in the lower troposphere there. I understand that part . Then, the authors follow the explanation saying that the wetter basic state over the equatorial Atlantic can lead to stronger local precipitation responses to NTA SST exciting, thereby, stronger subtropical teleconnection in relaying the Atlantic influences into the tropical Pacific. I really don` t understand this sentence and what do the authors want to state. Why does a wetter basic state over the equatorial leads to stronger responses to NTA ? there is no explanation about that.

Aso, Figure 5 is very difficult to understand and I think that there are some errors.. In the legend, it is indicated that negative precipitation goes with " + +" symbols (weird) ..also, it says that the winds promotes enhancing precipitation in the equator but in the figure there are " - " symbols over that region.

"North Atlantic Oscillation controls multidecadal changes in the North Tropical Atlantic-Pacific connection" (NCOMMS-22-28907) by Ding et al.

Responses to Reviewers #1 and #2

November 2, 2022

I. Reviewer #1

I-A. Response to general comments

This study provides a new explanation for the multidecadal changes in the North Tropical Atlantic (NTA)–ENSO connection, suggesting that the multidecadal variability associated the North Atlantic Oscillation (NAO) could principally control the multidecadal changes in the NTA–ENSO connection. The findings show that the phase of the NAO is key to understanding the impacts of the NTA SST on the tropical Pacific Ocean. This manuscript presents a novel view of topical Atlantic-Pacific inter-basin interactions, results of which have important implications for ENSO prediction and a better understanding of inter-basin interactions between the tropical Atlantic and Pacific. These results presented in this manuscript are interesting. This manuscript is well-written and the obtained results are reasonable. This manuscript is suggested to be accepted for publication in the journal after some revisions.

Response: We sincerely thank the reviewer for the positive comments and thoughtful review, which significantly help us to improve this manuscript. We have carefully revised the manuscript according to the comments and suggestions raised by the reviewer. Please find below a detailed point-by-point response to all comments (reviewers' comments in black, our replies in blue).

I-B: Response to major comments

1. *The springtime NTA SST may have important climatic effects in many aspects. As stated in the introduction of the manuscript, the NTA SST may play an important role in shaping the evolution of ENSO events and could therefore offer a potential precursor for ENSO. In addition to the potential precursor for ENSO, the springtime NTA SST has also been suggested to play important roles in modulating the western North Pacific (WNP) tropical cyclone (TC) genesis (Huo et al., 2015), western Pacific subtropical High (Wang et al., 2017; Chen et al., 2021), and the Indian Ocean Dipole (Zhang et al., 2022). Therefore, the springtime NTA SST may also offer a potential precursor for WNP TC numbers, the intensity of the western Pacific subtropical High, and the Indian Ocean Dipole. These associated information are suggested to be added in the manuscript, to further highlight the importance of NTA SST in climate variability and climate prediction.*

Response: We appreciate the reviewer's suggestions. In the light of the reviewer's suggestions, we have added the following sentences in the section “Introduction” of the revised manuscript (please see lines 62-65):

“Beyond the Atlantic sector, the NTA SST can exert pronounced impacts on the western North Pacific (WNP) tropical cyclone genesis (Huo et al., 2015), the WNP subtropical High (Chen et al., 2021), and the Indian Ocean Dipole (Zhang et al., 2022).”

Huo, L. W., Guo, P. W., Hameed, S. N., & Jin, D. C. The role of tropical Atlantic SST anomalies in modulating western North Pacific tropical cyclone genesis. *Geophys. Res. Lett.* 42, 2378–2384 (2015).

Chen, S. F., Chen, W., Wu, R. G., Yu, B., & Song, L. Y. Performance of the IPCC AR6 models in simulating the relation of the western North Pacific subtropical high to the spring northern tropical Atlantic SST. *Int. J. Climatol.* 41, 2189–2208 (2021).

Zhang, G. L., Wang, X., Xie, Q., Chen, J. P., & Chen, S. Strengthening impacts of spring sea surface temperature in the north tropical Atlantic on Indian Ocean dipole after the mid-1980s. *Clim. Dyn.* 59, 185–200 (2022).

2. *One recent study (Chen et al. 2022) proposed that the preceding winter North*

Pacific Oscillation (NPO) is an important source of the internal climate variability that modulates the spring NTA–ENSO connection. However, the observation data used in Chen et al. (2022) is from 1948 and is shorter than the data used in this study. It is suggested to further verify whether NPO or NAO control the multidecadal changes in the NTA–ENSO connection? Or both NPO and NAO may make contributions. Any linkage between the multidecadal variability of the NAO and the NPO?

Response: We are grateful for the reviewer’s helpful comments. As the reviewer mentioned, Chen et al. (2022) investigated the influences of the North Pacific Oscillation (NPO) on the relationship between spring NTA SST anomalies and the succedent winter ENSO for the period of 1979–2005. Their study mainly focused on the modulation effect of the NPO on the NTA–ENSO connection over interannual timescales. Over multidecadal timescales, the variability of the NPO is much weaker than that of the NAO (see Figure A1 below). There is a weak correlation between the NAO and NPO indices during boreal winter–spring seasons (DJFMAM) for the period of 1900–2021 ($R = 0.53$; not significant even at the 90% confidence level).

We further examined the relationship between the NTA–ENSO connection and NPO over multidecadal timescales. Multidecadal fluctuations of the NTA–ENSO connection are much more highly anti-correlated with those of the NAO ($R = -0.94$) than with those of the NPO ($R = -0.42$) (see Figure A2 below). This suggests that multidecadal NAO variability may dominate multidecadal fluctuations in the NTA–ENSO connection. However, despite the lack of significant correlation between multidecadal fluctuations of the NPO and NTA–ENSO connection over the entire period, we cannot exclude the possibility that the NPO may exert a modulation effect on the NTA–ENSO connection in a particular period (such as the 1913–1939 period; Figure A2). Therefore, we have added the following sentences in the section “Discussion” of the revised manuscript (please see lines 278-281):

“Chen et al. (2022) suggested that the North Pacific Oscillation (NPO) could also have a modulation influence on the NTA–ENSO connection. It is likely that the NAO and NPO may play a joint role in modulating the NTA–ENSO connection. Further investigations are required to better understand these two important issues.”

Chen, S. F., Chen, W., Yu, B., & Li, Z. B. Impact of internal climate variability on the relationship between spring northern tropical Atlantic SST anomalies and succedent winter ENSO: the role of the North Pacific Oscillation. *J. Clim.* 35, 537–559 (2022).

Figure A1. The 21-year running averages of the normalized NAO (red line) and NPO (blue line) indices during the boreal winter–spring seasons (DJFMAM) for the period 1900–2021.

Figure A2. Relationship between the NTA–ENSO connection and NAO/NPO in observations. (a) The 21-year sliding correlation coefficients between the boreal spring (MAM0) NTA SST index and subsequent boreal winter (D0JF1) Niño3.4 index (red line), and the 21-year running averages of the normalized NAO index during the previous boreal winter–spring seasons (D–1JFMAM0) (blue line) for the period 1900–2021 in observations. The correlation coefficient between the NTA–ENSO connection and NAO is -0.94 (significant at the 95% confidence level). The impact of the previous winter (D–1JF0) ENSO has been removed from the MAM0 NTA SST index using linear regression with respect to the Niño3.4 index. (b) Same as (a), but for the NTA–ENSO connection and NPO. The correlation coefficient between the NTA–ENSO connection and NPO is -0.42 (not significant even at the 90% confidence level).

3. *The changes in the NTA–ENSO connection may influence the ENSO properties largely. It is suggested to add more relevant discussions in this regard in the manuscript. Considering that the NTA-triggered ENSO tends to be the central Pacific (CP) type rather than the eastern Pacific (EP) type, the multidecadal changes in the NTA–ENSO connection may influence the dominant degree of EP and CP ENSO in different decades. In addition, the NTA SST could promote a faster ENSO phase transition, thus helping to shape the dominant period of the ENSO (Is the quasi-biennial (~2-3 yr) rhythm or the quadrennial (~4-5 yr) rhythm dominant?) As stated in the manuscript, the AMO-based linear model predictions suggest a likely weakening of the NTA-ENSO connection in the near future. How this weakening of the NTA-ENSO connection may influence the ENSO properties (the dominant type and the dominant period) in the near future? These points may further highlight the importance of the NTA–ENSO connection changes in the climate system.*

Response: We appreciate these constructive suggestions. As the reviewer mentioned, the NTA SST has been suggested to be closely related to the development of the CP El Niño (Ham et al. 2013). Moreover, the NTA SST has also been suggested to promote a faster ENSO phase transition, thus helping to shape the dominant period of ENSO (Ham et al. 2013; Wang et al. 2017). Our findings have suggested that the NAO could modulate multidecadal changes in the NTA–ENSO connection. Therefore, it is very likely that the NAO may influence the ENSO properties (including its spatial pattern and period) by modulating the NTA–ENSO connection. In the light of the reviewer's suggestions, we have added the following sentences in the section “Discussion” of the revised manuscript (please see lines 270-278):

“It has been well recognized that two types of El Niño exist in the tropical Pacific, characterized by larger SST anomalies over the eastern Pacific (EP) and central Pacific (CP), respectively. The NTA SST has been suggested to be closely related to the development of the CP El Niño. Moreover, it may also promote a faster ENSO phase transition, thus helping to shape the dominant period of ENSO. Our

findings further indicate that the NAO may influence the ENSO properties (including its spatial pattern and period) by modulating the NTA–ENSO connection. The weakening of the NTA–ENSO connection in the near future would likely lead to changes in ENSO properties.”

Ham, Y. G., Kug, J. S., Park, J. Y., & Jin, F. F. Sea surface temperature in the north tropical Atlantic as a trigger for El Niño/Southern Oscillation events. *Nat. Geosci.* 6, 112–116 (2013).

Wang, L., Yu, J. Y., & Paek, H. Enhanced biennial variability in the Pacific due to Atlantic capacitor effect. *Nat. Commun.* 8, 14887 (2017).

4. *Lines 177-180: "The intensified northeasterly trade winds can enhance the wind-evaporation-SST (WES) feedback mechanism, which acts to increase the persistence of the NTA SST anomalies. " Here, could the enhancement of the WES feedback mechanism be supported by the observational data? Why the intensified northeasterly trade winds can enhance the WES feedback? Why these mentioned changes could act to increase the persistence of the NTA SST anomalies? Any local air-sea positive feedback is included in the process?*

Response: Thanks for the insightful comments from the reviewer. Following Czaja et al. (2002) and Vimont et al. (2009), the sensitivity of latent heat flux to zonal winds, or the WES parameter (WESp) can be expressed by the following relationship:

$$\text{WESp} = -\frac{\partial \text{LH}}{\partial u} = -\text{LH} \frac{u}{W^2}, \quad (1)$$

where LH is latent heat flux, u is the 10 m zonal wind, and W is the total wind speed. The total wind speed can be decomposed as:

$$W = \sqrt{u^2 + v^2 + w_*^2}, \quad (2)$$

where w_* is turbulent background wind speed. Since zonal wind dominates the WES feedback, Equation (1) can be simply expressed by:

$$\text{WESp} \sim \frac{1}{1 + \left(\frac{w^*}{u}\right)^2} . \quad (3)$$

Equation (3) indicates that a stronger mean zonal wind generally corresponds to a more intense WES feedback, and vice versa. The intensified northeasterly trade winds are associated with a stronger mean zonal wind. Therefore, we argue that the intensified northeasterly trade winds tend to enhance the WES feedback mechanism, which is supported by the findings of Yu et al. (2015). To clearly explain how the intensified northeasterly trade winds tend to enhance the WES feedback, these descriptions of the WES parameter have been added to the section “Methods” of the revised manuscript (please see lines 332-345).

Based on the NCEP/NCAR reanalysis data, we compared the WES parameter (WESp) over two periods (1961–1985 and 1992–2016, respectively), which correspond to negative and positive phases of the NAO. Over most of the NTA region (0°–15°N, 90°W–20°E), there is an increase of the WES parameter in the positive phase of the NAO than in the negative phase of the NAO (see Figure A3 below). According to Equation (3), this increase in the WES parameter may have contributions from the intensified northeasterly trade winds in the positive phase of the NAO. The increased WES feedback tends to enhance the coupling among SST, winds, and evaporation in the positive phase of the NAO. This increased local air-sea positive feedback may help sustain the NTA SST anomalies during boreal spring and therefore enhance the persistence of the NTA SST anomalies from spring to summer.

Czaja A, Van der Vaart P, Marshall J (2002) A diagnostic study of the role of remote forcing in tropical Atlantic variability. *J Clim* 15(22):3280–3290

Vimont DJ, Alexander M, Fontaine A (2009) Midlatitude excitation of tropical variability in the Pacific: the role of thermodynamic coupling and seasonality. *J Clim* 22(3):518–534

Yu, J. Y., Kao, P. K., Paek, H., Hsu, H. H., Hung, C. W., Lu, M. M., & An, S. I. Linking emergence of the central Pacific El Niño to the Atlantic multidecadal oscillation. *J. Clim.*

Figure A3. The WES parameter (WESp ; W s m^{-3}) over (a) the 1961–1985 and (b) 1992–2016 periods, and (c) the difference between (b) and (a).

I-C: Response to minor comments

1. *Line 248: “inter-basin interactions between the Atlantic”. Some words are missing here: inter-basin interactions between the Atlantic and the Pacific ?*

Response: We thank the reviewer for pointing this out. We have completed the sentence as follows:

“These findings have important implications for ENSO prediction and a better understanding of inter-basin interactions between the Atlantic and Pacific Oceans.”

2. *Line 24: Nature Geoscience should be Nature Communications.*

Response: We thank the reviewer for pointing this out. We have corrected this mistake in the revised manuscript.

3. *For the title, "the multidecadal changes" need to be included? Such as, North Atlantic Oscillation controls the multidecadal changes in the North Tropical Atlantic-Pacific connection.*

Response: We appreciate the good suggestion from the reviewer. Following the reviewer's suggestion, the title has been changed to “North Atlantic Oscillation controls multidecadal changes in the North Tropical Atlantic-Pacific connection”.

II. Reviewer #2

II-A. Response to general comments

The manuscript “North Atlantic Oscillation controls the North Tropical Atlantic-Pacific connection” by Ruiqiang Ding et al deals with the north tropical Atlantic impact on ENSO and its non-stationarity. The authors suggest that the decadal variability of the North Atlantic Oscillation acts as modulator of this connection, being during the positive phase of NAO when the connection works.

The authors indicate that, although the positive phase of AMO coincides with enhancement of NTA-ENSO, there is almost no correlation between the correlation curve NTA-ENSO and AMO but yes with that of the NAO .

Although the experimental set up is fine and also the results seem to be promising, I have some concerns about the explanations given in the paper and, for this reason, I cannot recommend this paper to be published in its present form in this high impact journal.

Response: We appreciate all the thoughtful comments and constructive suggestions from the reviewer. We have carefully modified the manuscript according to the comments and suggestions raised by the reviewer. Please see our point-by-point responses (in blue) to the comments raised by the reviewer below.

II-B: Response to specific comments

1. *The linear model based on AMO implies that positive AMO produces, 20 year later a positive NAO. What is the mechanism behind? On the one hand, the authors use a linear model to predict NAO as a function of AMV , but NAO can be internally. Regardless of that, I am concern about the signs of the relations found. For example, Sutton et al (2019) indicates, in its figure 8, how a positive AMV phase associated with a stronger AMOC leads to a negative winter NAO response (i.e., weaker subtropical high and subpolar low over the North Atlantic) , so NAO- is related to positive AMV. Nevertheless, in the present manuscript the authors*

obtain a positive NAO with NTAENSO relation, which is not in agreement with the results from Wang et al (2017) in which the authors relate NTA-ENSO with positive AMV. Even if NAO can be predicted from AMV, the sign of the relation does not agree with Wang et al (2017) results. Please clarify this.

Response: Thanks for the insightful comments from the reviewer. Lead-lag correlation between the AMO and NAO indicates that the negative AMO leads the positive NAO by ~20 years (i.e., AMO⁻→NAO⁺) (see Figure B1 below). On the physical mechanism behind this lead-lag relationship between the AMO and NAO, Sun et al. (2015) proposed a delayed oscillator framework to explain the interactions between multidecadal NAO and AMO/AMOC and the associated quasi-periodic multidecadal oscillation. The positive NAO corresponds to a strengthening of the Icelandic low and Azores high, an increase in the atmospheric forcing of the ocean. The accumulated effect of the NAO atmospheric forcing causes multidecadal variations of the AMOC and AMO SST anomalies, with the NAO leading AMO/AMOC by around two decades. The large inertial of the enhanced AMOC causes an increase in ocean heat transport to the subpolar/polar region, leading to subpolar warming, decreasing in the meridional temperature gradient, and westerlies weakening, providing delayed negative feedback from AMO/AMOC to the NAO. Moreover, this delayed negative feedback results from the delayed response of the North Atlantic tripole (NAT) to AMOC-induced heat transport, characterized by warming in the subpolar and cooling in the mid-latitudes. The time delay for the negative feedback can be further explained physically, interpreted as the time it takes for transport along the AMOC-related oceanic heat transport pathway. Based on the maximum of the overturning streamfunction (around 16 Sv) and the basin scale of the North Atlantic Ocean, the theoretical time delay of NAT relative to the AMO is estimated as ~20 years, consistent with the timescale of overturning advection (Winton and Sarachik 1993; Ou 2011). Therefore, the physical mechanism for the delayed negative feedback of AMO on the NAO is through the overturning advection process and the changes

in the surface temperature meridional gradient, which strengthens/weakens the westerlies and leads to the changes in the NAO phase.

The feedback of NAT SST anomalies on the NAO presented above is consistent with previous studies, which have also shown that the atmospheric responses to the tripole-like SST anomalies strongly project onto the NAO (e.g., Sutton et al. 2019; Zhang et al. 2019). These studies have been referenced in the revised manuscript. Given the feedback between the AMO and NAO, it is feasible to use the AMO to predict the NAO over multidecadal timescales.

Using long records of reanalysis data, we show that there is almost no simultaneous correlation between the AMO and multidecadal fluctuations of the NTA–ENSO connection over the entire period ($R = -0.15$; see Figure B2a below). Instead, lead-lag correlations between the AMO and NTA–ENSO connection curves indicate that changes in the AMO lead those of the NTA–ENSO connection by ~20 years (see Figure B3a below). In contrast, there is a simultaneous correlation between the NAO and multidecadal fluctuations of the NTA–ENSO connection (see Figure B3b below). These results suggest that the NAO could exert a direct modulation effect on the NTA–ENSO connection, while the AMO may exert an indirect (delayed) effect on the NTA–ENSO connection through its interaction with the NAO. A systematic cause-and-effect relationship among the AMO, NAO, and NTA–ENSO connection may exist as follows:

As the reviewer mentioned, our results differ from those of Wang et al. (2017) who related the AMO+ to the enhanced NTA–ENSO connection. We note that the study by Wang et al. (2017) only utilized observational data from 1948 onwards. This period is too short and includes only one negative phase and one positive phase of the AMO. The positive phase of the AMO almost coincides with the enhanced

NTA–ENSO connection after the late 1990s (see Figure B2a below). Therefore, Wang et al. (2017) attributed the enhanced NTA–ENSO connection after the late 1990s to the positive phase of the AMO. However, if using data extending back to 1900, there is almost no simultaneous correlation between the AMO and multidecadal fluctuations of the NTA–ENSO connection (see Figure B2a below). This implies that the relationship between the AMO+ and enhanced NTA–ENSO connection was not supported by the observational data. It seems that the coincidence between the AMO+ and enhanced NTA–ENSO connection after the late 1990s may arise from the oscillation of the AMO itself (from the AMO– to the AMO+, and the AMO– leads the AMO+ by ~20 yrs).

According to our analysis, the AMO- leads the NAO+ by ~20 yrs; the latter would subsequently lead to the enhanced NTA–ENSO connection. Therefore, we conclude that the NAO could exert a direct modulation effect on the NTA–ENSO connection. Our results are supported by long records of observational data and numerical experiments. It should be emphasized that our results do not deny the effects of the AMO on the NTA–ENSO relationship, but rather point out that the AMO may have an indirect (delayed) effect on the NTA–ENSO relationship through its interaction with the NAO.

Sun, C., J. Li and F. F. Jin, 2015: A delayed oscillator model for the quasi-periodic multidecadal variability of the NAO. *Climate Dynamics*, 45, 2083-2099.

Winton M, Sarachik ES (1993) Thermohaline oscillations induced by strong steady salinity forcing of ocean general-circulation models. *J Phys Oceanogr* 23:1389–1410

Ou H-W (2011) A minimal model of the Atlantic multidecadal variability: its genesis and predictability. *Clim Dyn* 38:775–794

R. T. Sutton, G. D. McCarthy, J. Robson, B. Sinha, A. T. Archibald, and L. J. Gray, Atlantic multidecadal variability and the U.K. ACSIS program, *Bull. Am. Meteorol. Soc.* 99, 415–425.

Zhang, R., Sutton, R., Gokhan, D., Kwon, Y.-O., Marsh, R., Yeager, S. G., Amrhein, D. E., and Little, C. M. A review of the role of the Atlantic Meridional Overturning Circulation

in Atlantic multidecadal variability and associated climate impacts. *Rev. Geophys.* 57, 316–375 (2019).

Figure B1. Lead-lagged correlation between the DJFMAM-averaged NAO and AMO indices (1900–2021). The NAO and AMO indices are smoothed with a 21-year running mean. The dashed lines denote the 95% confidence levels.

Figure B2. (a) The 21-year sliding correlation coefficients between the boreal spring (MAM0) NTA SST index and subsequent boreal winter (D0JF1) Niño3.4 index (red line), and the 21-year running averages of the normalized AMO index during the previous boreal winter–spring seasons (D-1JFMAM0) (blue line) for the period 1900–2021 in observations. (b) The 21-year sliding correlation coefficients between the boreal spring (MAM0) NTA SST index and subsequent boreal winter (D0JF1) Niño3.4 index (red line; left axis), and the 21-year running averages of the normalized NAO index during the previous boreal winter–spring seasons (D-1JFMAM0) (blue line; right axis).

Figure B3. (a) Lead-lagged correlations between the DJFMAM-averaged AMO and NTA–ENSO connection curves in Figure B2a. The dashed lines denote the 95% confidence levels. (b) Same as (a), but for lead-lagged correlations between the DJFMAM-averaged NAO and NTA–ENSO connection curves in Figure B2b.

2. *Also, there are some parts of the text that are very difficult to read. For example, the authors claim that the positive NAO strengthen the northeasterly winds promoting low level convergence over the equatorial Atlantic and enhancing precipitation and relative humidity in the lower troposphere there. I understand that part. Then, the authors follow the explanation saying that the wetter basic state over the equatorial Atlantic can lead to stronger local precipitation responses to NTA SST exciting, thereby, stronger subtropical teleconnection in relaying the Atlantic influences into the tropical Pacific. I really don't understand this sentence and what do the authors want to state. Why does a wetter basic state over the equatorial leads to stronger responses to NTA ? there is no explanation about that.*

Response: Thanks for the constructive comments from the reviewer. According to the comments from the reviewer, we provide further explanation of how the wetter basic state over the equatorial Atlantic can lead to stronger local precipitation responses to NTA SST anomalies:

From the moisture budget perspective, on the interannual variability, tropical precipitation anomalies are mainly dominated by the moisture convergence induced by anomalous circulation (Neelin 2007):

$$P' \sim -\langle \bar{q} * \nabla \cdot \bar{\mathbf{u}}' \rangle, \quad (1)$$

where P denotes the tropical precipitation, the prime denotes interannual variability, \bar{q} denotes the background mean-state specific humidity, and $\nabla \cdot \bar{\mathbf{u}}'$ denotes the moisture convergence induced by anomalous circulation, respectively. $\langle \rangle$ denotes vertical integration. For the same magnitude of anomalous convergence induced by local SSTA forcing, the precipitation response would be stronger when the mean-state low-level moisture were more. Furthermore, when the precipitation anomalies are generated, they tend to be further amplified by internal positive feedback between convection and cloud radiative forcing in the tropical atmosphere.

Thus, the wetter basic state over the equatorial Atlantic in the positive phase of the NAO can lead to stronger local precipitation responses to NTA SST anomalies, thereby exciting stronger subtropical teleconnections that are more effective in relaying the Atlantic influences into the tropical Pacific. Other studies that evaluated the influences of the NTA SST on the ENSO variability with the CMIP3/5 models also found the important role of the mean precipitation in determining the strength of the tropical Pacific response to the NTA SST (Ham et al. 2015).

To clearly explain how the wetter basic state over the equatorial Atlantic in the

positive phase of the NAO can lead to stronger local precipitation responses to NTA SST anomalies, the moisture budget analysis presented above has been added to the section “Methods” of the revised manuscript (please see lines 319-330).

Neelin, J. D., 2007: Moist dynamics of tropical convection zones in monsoons, teleconnections, and global warming. *The Global Circulation of the Atmosphere*, T. Schneider and A. Sobel Eds, Princeton University Press, 263–301.

Watanabe, M., Chikira, M., Imada, Y., & Kimoto, M. Convective control of ENSO simulated in MIROC. *J. Clim.* 24, 543–562 (2011).

Xiang, B., Wang, B., Yu, W., & Xu, S. How can anomalous western north Pacific subtropical high intensify in late summer? *Geophys. Res. Lett.* 40, 2349–2354 (2013).

Ham, Y. G., & Kug, J. S. Role of north tropical atlantic SST on the ENSO simulated using CMIP3 and CMIP5 models. *Clim. Dyn.* 45, 3103–3117 (2015).

3. *Also, Figure 5 is very difficult to understand and I think that there are some errors.. In the legend, it is indicated that negative precipitation goes with “ + +” symbols (weird) ..also, it says that the winds promotes enhancing precipitation in the equator but in the figure there are “ - “ symbols over that region.*

Response: We thank the reviewer for pointing this out. We have modified Figure 5 to address the reviewer’s comment. In the revised figure, positive precipitation anomalies are indicated by “+ +” symbols, and negative precipitation anomalies are indicated by “● ●” symbols (see Figure B4 below). We have also tried other different symbols and the current ones look a bit better.

We would like to clarify that the intensified northeasterly trade winds associated with the positive NAO enhance **background mean-state precipitation** over the equatorial Atlantic during boreal spring. The wetter basic state would lead to a stronger **negative precipitation anomaly response** to a negative NTA SST anomaly (SST–), thereby exciting stronger subtropical teleconnections that more

readily relay the Atlantic influences into the tropical Pacific. Here the negative precipitation anomaly over the equatorial Atlantic is a response to a negative NTA SST anomaly, rather than the background mean-state precipitation. To demonstrate this point more clearly, we have revised the descriptions of the figure (see the caption of Figure B4 below).

Figure B4. Schematic representation of two major mechanisms behind the NAO modulation of the NTA–ENSO connection. (a) NAO modulation of background mean-state precipitation over the equatorial Atlantic. The intensified northeasterly trade winds associated with the positive NAO favor stronger background mean-state precipitation over the equatorial Atlantic during boreal spring. The wetter basic state would lead to a stronger negative precipitation anomaly response to a negative NTA SST anomaly (SST–), thereby exciting stronger subtropical teleconnections that more readily relay the Atlantic influences into the tropical Pacific. (b) NAO modulation effect on NTA SST persistence. The intensified northeasterly trade winds associated with the positive NAO deepen the ocean mixed layer and enhance the WES feedback mechanism, which in turn increase the persistence of the NTA SST anomalies from spring to summer, thereby generating more persistent subtropical teleconnections that continuously

relay the Atlantic influences into the tropical Pacific. Colors denote the seasons when significant anomalies occur. Crosses and circles denote the locations of positive and negative precipitation anomalies, respectively.

REVIEWERS' COMMENTS

Reviewer #1 (Remarks to the Author):

The authors have revised the manuscript and responded to the previous comments. In the revised manuscript, the authors have added more related references to further highlight the importance of the North Tropical Atlantic (NTA) SST in climate variability and climate prediction, in modulating the western North Pacific tropical cyclone genesis, western Pacific subtropical High, and the Indian Ocean Dipole. Discussions on the possible influences from the NPO have been added in the revised manuscript. The WES parameter (WESp) has been used to diagnose the WES mechanism. Also, more discussions about possible influences on the near future ENSO properties (including its spatial pattern and period) have been added. This manuscript presents a novel view of tropical Atlantic-Pacific inter-basin interactions, results of which have important implications for ENSO prediction and a better understanding of inter-basin interactions between the tropical Atlantic and Pacific.

I am satisfied with the revisions and the responses. The manuscript is suggested to be accepted for publication in the journal.

**"North Atlantic Oscillation controls multidecadal changes in the
North Tropical Atlantic–Pacific connection"
(NCOMMS-22-28907A) by Ding et al.
Responses to Reviewers**

February 2, 2023

REVIEWERS' COMMENTS

Reviewer #1 (Remarks to the Author):

The authors have revised the manuscript and responded to the previous comments. In the revised manuscript, the authors have added more related references to further highlight the importance of the North Tropical Atlantic (NTA) SST in climate variability and climate prediction, in modulating the western North Pacific tropical cyclone genesis, western Pacific subtropical High, and the Indian Ocean Dipole. Discussions on the possible influences from the NPO have been added in the revised manuscript. The WES parameter (WESp) has been used to diagnose the WES mechanism. Also, more discussions about possible influences on the near future ENSO properties (including its spatial pattern and period) have been added. This manuscript presents a novel view of tropical Atlantic-Pacific inter-basin interactions, results of which have important implications for ENSO prediction and a better understanding of inter-basin interactions between the tropical Atlantic and Pacific.

I am satisfied with the revisions and the responses. The manuscript is suggested to be accepted for publication in the journal.

Response: We would like to thank the reviewers for their valuable time and effort in reviewing our manuscript. We sincerely appreciate all helpful comments and suggestions raised by the reviewers, which

help us to improve the quality of our manuscripts.